# Cost analysis of photobiomodulation in tibia fracture in the Brazilian public health system

**Frederico Carlos Jana Neto**[1,2,3], **Ana Luiza Cabrera Martimbianco**[4,5], **Diogo Valvano de Medeiros**[3], **Fernanda Carolina Felix**[3], **Raquel Agnelli Mesquita-Ferrari**[1,6], **Sandra Kalil Bussadori**[1,6], **Cinthya Cosme Gutierrez Duran**[1,7], **Lara Jansiski Motta**[1], **Estela Capelas Barbosa**[8]*, **Kristianne Porta Santos Fernandes**[1,7]

**1** Postgraduate Program in Biophotonics Applied to Health Sciences, Universidade Nove de Julho (UNINOVE), São Paulo, SP, Brazil, **2** Orthopedics and Traumatology Group Conjunto Hospitalar do Mandaqui, São Paulo, SP, Brazil, **3** Medicine School Universidade Nove de Julho (UNINOVE), São Paulo, SP, Brazil, **4** Postgraduate Program in Health and Environment. Universidade Metropolitana de Santos (UNIMES), Santos, SP, Brazil, **5** Health Technology Assessment Center, Hospital Sírio-Libanês (NATS-HSL), São Paulo, SP, Brazil, **6** Postgraduate Program in Rehabilitation Sciences, Universidade Nove de Julho (UNINOVE), São Paulo, SP, Brazil, **7** Postgraduate Program in Medicine, Universidade Nove de Julho (UNINOVE), São Paulo, SP, Brazil, **8** Population Health Sciences, Bristol Medical School, Bristol University, Bristol, United Kingdom

* e.capelasbarbosa@bristol.ac.uk

**Data Availability Statement:** The manuscript contains all data defined by PLOS ONE to be the 'minimum dataset', including means, standard

## Abstract

Managing tibial fractures requires substantial health resources, which costs the health system. This study aimed to describe the costs of photobiomodulation (PBM) with LEDs in the healing process of soft tissue lesions associated with tibial fracture compared to a placebo. Economic analysis was performed based on a randomized controlled clinical trial, with a simulation of the cost-effectiveness and incremental cost model. Adults (n = 27) hospitalized with tibia fracture awaiting definitive surgery were randomized into two distinct groups: the PBM Group (n = 13) and the Control Group with simulated phototherapy (n = 14). To simulate the cost-effectiveness and incremental cost model, the outcome was the evolution of wound resolution by the BATES-JENSEN scale and time of wound resolution in days. The total cost of treatment for the Control group was R$21,164.56, and a difference of R$7,527.10 more was observed when compared to the treatment of the PBM group. The proposed intervention did not present incremental cost since the difference in the costs to reduce measures between the groups was smaller for the PBM group. When analyzing the ICER (Incremental cost-effectiveness ratio), it would be possible to save R$3,500.98 with PBM and decrease by 2.15 points in the daily average on the BATES-JENSEN scale. It is concluded, therefore, that PBM can be a supportive therapy of clinical and economic interest in a hospital setting.

## Background

Traffic injuries are the leading cause of death among young people worldwide. Tibial fractures are the most common long bone fractures and essential to hospital operating room

deviations and other measures. The collected data is stored in the RedCap repository (https://www.redcaphospitalmandaqui.com/). The data, which has been kept anonymous regarding the participants, follows the guidelines of the Ethics Committee of Conjunto Hospitalar do Mandaqui (CHM) (protocol code 3.946.372, March 31, 2020), located at Rua Voluntários da Pátria, 4301 - Mandaqui - São Paulo - Brazil. Phone: +55 11 2281-5174. Email: cepchm@gmail.com."

**Funding:** KPSF, SKB and RAMF were supported by CNPq - National Council for Technological and Scientific Development (CNPq, grant n. 304330/2020-5, n. 306577/2020-8 and n.310491/2021-5). The sponsors had no role in the design, execution, interpretation, or writing of the study.

**Competing interests:** The authors have declared that no competing interests exist.

procedures. Patients remain for many days in health facilities, which imposes a cost to the health system [1]. These fractures are susceptible to becoming exposed fractures [1] In the United Kingdom, the incidence is 5.6 per 100,000 people per year and is often associated mainly with infection, pseudarthrosis and amputation in low- and middle-income countries [2]. Under these conditions, patients stay longer in hospital facilities, which assigns costs to the health system [1].

The National Center for Health Statistics (NCHS) reports an annual incidence of 492,000 tibial and fibular fractures annually in the United States. Patients with tibia fractures remain in the hospital for 569,000 days of hospitalization and require 825,000 medical consultations annually in the country [1, 2].

In Brazil, it is estimated that 11,000 surgeries are performed for the treatment of tibial fracture by the Unified Health System (SUS) for R$9,317,006.85 for the system [3].

In the United Kingdom, the prevalence of complex wounds is estimated at 14.7 per 10,000 inhabitants per year, representing a universe of 80,000 patients at three billion pounds per year [4]. In the United States of America, it is estimated that 2% of the population is affected annually by complex wounds, which represents a cost of 25 billion dollars per year [5, 6].

A 2021 systematic review [2] of 34 studies on the economic impact of tibia fractures found significant cost variations across countries. In the United States, initial hospital costs ranged from £5,705 to £126,479, and in the United Kingdom, they ranged from £9,401 to £13,855. Other countries, including Canada, Singapore, Switzerland, Belgium, and Denmark, were also studied. Notably, one low-income country had considerably lower costs, ranging from £356 to £1,069.

Overall, the review indicated that initial hospital costs for reconstruction procedures were higher than for controls, while costs for amputations were lower. Infection cases incurred higher costs compared to non-infected cases. The average hospital stay across 15 studies was 56 days, with an average recovery time of 50 weeks. The review emphasized the impact of tibia fractures on patients' ability to work, with 40% experiencing partial or complete work impairment. In conclusion, the review underscored the need for further research into the economic impact of tibia fractures in diverse settings, particularly in low- and middle-income countries [2]. It also highlighted the importance of conducting economic analyses in the Brazilian context to develop standardized cost assessment tools and gain a more comprehensive understanding of tibia fracture-related costs in this region.

Health Technology Assessment (HTA) is crucial for making decisions about new drugs and healthcare technologies in different healthcare systems. It primarily considers a balance between the clinical benefits for patients and the economic costs associated with introducing these new technologies into the healthcare system. While economics in general deals with allocating limited resources among competing needs, health economics specifically focuses on allocating resources to enhance health. It's a branch of economics that examines efficiency, effectiveness, and the value of resources in healthcare [7, 8].

Economic evaluations have various potential uses, including creating public reimbursement lists, negotiating prices, developing clinical practice guidelines, and communicating with healthcare providers. These evaluations are vital for understanding the economic aspects of health and disease and recognizing the barriers to accessing adequate healthcare [7–10].

The management of health services constantly seeks strategies to contain health costs and to reduce the possibilities of infection, amputation, and prolonged hospitalization time in wound resolution; new adjuvant and complementary procedures, such as photonic therapies, have been clinically tested.

Photon therapies apply light in different clinical situations for prevention and health care. The applications involve high-intensity sources for use in surgeries, photodynamic treatment

and photobiomodulation that have advanced with studies that show inhibitory or excitatory effects on tissues and organisms depending on the dosimetric parameters used [11, 12].

Photobiomodulation is a therapy using a laser or LED light that has been indicated for pain relief and tissue repair. Low-intensity light irradiation causes non-thermal, non-ionizing effects mediated by photochemical reactions. Therapy is described in the literature to treat damage to sports injuries and musculoskeletal disorders; it is applicable to reduce pain and scar size after surgery. Photobiomodulation may be associated with improvements in pain and physical function in treating bone fractures [13–16].

The application of light stimulates the respiratory cycle in mitochondria and increases adenosine triphosphate molecules[11] that reduce swelling and pain [11, 12, 17]. A study with laser light application at a wavelength of 830 nanometers helped treat a tibial stress fracture. The group treated with laser presented an earlier resolution of symptoms and painless ambulation with less recurrence [18]. The results of a randomized clinical trial suggest that LED photobiomodulation is a promising treatment for traumatic soft tissue injuries associated with lower limb fractures as it demonstrated efficacy and safety in cases of soft tissue injuries associated with fractures in the lower limbs, reducing the time needed for definitive surgery and hospitalization period [19].

Considering the possibility of inserting photon therapies in hospitals to reduce the days of hospitalization, this study aims to describe the costs of treatment using LEDs in the healing process of soft tissue injuries associated with tibial fracture compared to placebo.

## Methods

This economic evaluation was made from the point of view of the Brazilian public health system. All costs are expressed in reais for the financial year 2022. A comprehensive economic perspective would be desirable. However, due to photonic therapies' emerging and recent characteristics, we have not yet found many controlled clinical studies of applying the technique in tibial fractures. Therefore, it was decided to perform the economic analysis based on a randomized controlled clinical trial in a single centre and simulate the cost-effectiveness and incremental cost model. The clinical results of this study have already been published [19].

The clinical study enrolled 27 adult patients, aged 18 to 72 years, who were admitted to the hospital with tibial fractures and were waiting to resolve soft tissue injuries before undergoing definitive surgery. They received multidisciplinary care following the standard protocol based on Advanced Trauma Life Support criteria [20]. Daily dressing changed using rayon dressings soaked in sterile petroleum jelly and covered with gauze and bandages. Patients were considered eligible for the study only after placing the external fixator, initial cleaning, and debridement in the operating room to ensure the presence of wounds that could not be closed immediately and required further treatment [19].

The participants were randomly assigned to either the PBM group (n = 13; using a 144 LED emitting diode device with wavelengths of 420nm, 660nm, and 850nm, 3J per point, for 10 minutes) or the simulated photon therapy (Control) group (n = 14; using a device with identical external features but without light transmission). The research was carried out at the Conjunto Hospitalar do Mandaqui (CHM) in São Paulo, Brazil, following the Declaration of Helsinki and approved by the Research Ethics Committee (3.946.372/CHM).

The inclusion criteria for the participant's selection involved adult individuals (over 18 years), hospitalized due to a traumatic lower limb soft-tissue injury, unfeasible for primary closure or definitive treatment of initial care injuries associated with a tibial and/or ankle fractures. Individuals who presented or reported: chronic systemic diseases, allergy to cefazolin and gentamicin, uncontrollable active bleeding, occlusive arteriopathies, neurovascular injury with a sensory deficit at the injury site, previous surgeries on the affected limb, local or

systemic changes that contraindicate surgical intervention, smoking, photosensitivity history, neurological and psychiatric disorders, use of anti-inflammatory drugs in the last 15 days before the trauma, and pregnant women were excluded. In addition, during treatment, individuals who presented any complication (bleeding, operative difficulty, neurovascular injury not diagnosed at admission, among others) at any stage of treatment were not considered or withdrawn from the study. Data from these participants would not be included in the statistical analysis but would be described and discussed as possible adverse events [19].

The assessments of the target outcomes were conducted by the field team in charge of the daily medical check-up (blinded to the group to which the participant was assigned), both before and daily throughout the intervention period until the wounds were determined to be resolved, i.e., with healthy granulation tissue present, free from necrosis or purulent discharge, and therefore suitable for primary closure, closure using a flap or graft, or for choosing secondary intention healing which marked the end of the protocol [19].

The clinical outcomes of this randomized trial have shown the effect and safety of multi-wavelength photobiomodulation in patients with soft tissue injuries associated with lower limb fractures, reducing the hospital stay [19]. These results indicate that multiwavelength photobiomodulation using LED is a promising therapy for traumatic soft tissue injuries associated with lower limb fractures.

To simulate the model of cost-effectiveness and incremental cost, the outcome was the evolution of wound resolution by the BATES-JENSEN scale and time of wound resolution in days. The Bates Jensen Wound Assessment Tool (BATES-JENSEN scale) was applied to all participants at the time of enrollment in the study and the days until they were ready for definitive surgery. The scale ranges from 9 to 65 points, with 9 points corresponding to the healed wound, 13 to the regenerating wound and scores above 60 points corresponding to tissue degeneration [21].

This study estimated direct medical costs, non-medical direct costs, and lost productivity.

The values were raised from the perspective of the Unified Health System (SUS) as a buyer of the service. For this purpose, the direct costs of the procedures performed during the research period were considered. The values of the materials used, professional fees, hospital costs and costs with the loss of productivity during the absence from work were computed. The reference sources of costs in Brazilian currency (Real) were the databases ComprasNet (Brazil's government procurement system), SIGTAP (Table of Procedures of the Unified Health System of Brazil), and the Brazilian Ministry of Health Price Database. The estimated value of the photon therapy device was R$800.00.

As this is the first clinical trial conducted with photobiomodulation in this clinical condition, we do not have the cost of the procedure in the Brazilian healthcare system's table. Therefore, to adhere to the methodology of health economic analysis, since the only modification between the PBM group and sham was the application of light, we considered the cost of the intervention as the cost of the equipment. The equipment costs were added to the costs of the experimental group. Since the only modification between the PBM and sham groups was the application of light [22].

The means of the variables of pain evaluation, BATES-JENSEN scale, and wound area were compared between the two groups by analysis of variance (ANOVA), adjusted by the baseline data and possible confounding variables. The cost and effect analysis approach was conducted by examining the ratio of cost differences and differences in intervention effects. Following the formula Cost (treatment 2)—Cost (treatment 1) / Effectiveness (treatment 2)—Effectiveness (treatment 1) [22]. The significance level established was 0.05. The analyses were performed in SPSS v.25 Statistica 12, SAS JMP® v.11 and Origin Pro 2019. The clinical data that support the results are stored on the RedCap platform.

## Results

The demographic and baseline data were published by the same group in their randomized clinical trial in 2023 [19]. When analyzing the data related to tibia injury, it was observed that there was no difference between the groups in terms of left or right-side involvement. In the PBM group, 38.5% (n = 5) had right-sided injuries, and 61.5% (n = 8) had left-sided injuries, while in the control group, 57.1% (n = 8) had left-sided injuries, and 42.9% (n = 6) had right-sided injuries. As for the type of trauma in both groups, the most common cause (PBM = 30.8% and Control = 42.9%) was motorcycle accidents, followed by falls (PBM = 23.1% and Control = 14.3%) and car accidents (PBM = 15.4% and Control = 14.3%). The initial wound area was also similar between the groups (PBM = 23.1% up to 10 cm2; 23.1% between 10 and 20 cm2, and 53.8% larger than 20 cm2, and Control = 28.6% up to 10 cm2; 28.6% between 10 and 20 cm2, and 42.9% larger than 20 cm2). No significant differences were found between groups regarding demographic characteristics, clinical conditions and severity of traumatic injuries [19].

Table 1 represents the calculations considering the average daily rates until resolution in each group: control (23.1 days) and PBM (13.1 days). Statistically significant differences were observed between the groups in the daily mean evaluation of wounds on the BATES-JENSEN scale. Groups in the total score (Control 34.26 x PBM 32.10); and in the items size (Control 2.51 x PBM 2.91); type of necrotic tissue (Control 3.33 x PBM 2.17); the amount of necrotic tissue (Control 2.47 x PBM 1.65); skin colour around the wound (Control 2.39 x PBM 1.64) and granulation tissue (Control 3.85 x PBM 3.54) (Table 2).

**Table 1. Treatment direct and indirect cost estimates.**

| Costs | Estimated Value (BRL) | Group Control | PBM Group |
|---|---|---|---|
| | | Total (BRL) | Total (BRL) |
| Hospitalization daily public network ward | 206.87 | 4,778.70 | 2,710.00 |
| External fixator | 340.00 | 340.00 | 340.00 |
| Orthopedic care with temporary immobilization | 13.00 | 13.00 | 13.00 |
| Emergency room fee | 119.58 | 119.58 | 119.58 |
| Gypsum room fee | 70.82 | 70.82 | 70.82 |
| Surgical arch/image intensifier (use) | 1,034.38 | 1,034.38 | 1,034.38 |
| Electric perforator for surgery (use) | 43.57 | 43.57 | 43.57 |
| Vacuum cleaner (use) | 77.58 | 77.58 | 77.58 |
| Oximeter (use) | 38.79 | 38.79 | 38.79 |
| Multifunction monitor (time) | 103.44 | 103.44 | 103.44 |
| Anesthesia trolley (use) | 35.63 | 35.63 | 35.63 |
| Dipyrone Monohydrate 1g 4 Tablets | 3.53 | 3.53 | 3.53 |
| Tramadol hydrochloride 50mg in 3 capsules | 5.43 | 5.43 | 5.43 |
| Cephalexin 500mg—10 Tablets | 13.52 | 13.52 | 13.52 |
| Rifocina 20ml Spray | 29.69 | 29.69 | 29.69 |
| Sterile gauze 13 threads—bested | 0.55 | 127.05 | 72,05 |
| Crepe bandage 10cm x 1.8mt | 7.59 | 1,753.29 | 994.29 |
| Change of dressing at the hospital | 38.79 | 8,960.49 | 5,081.49 |
| Daily rate of worker on leave (indirect cost) | 81.96 | 1,893.28 | 1,073.68 |
| Sick pay (estimated indirect cost) | 74.58 | 1,722.80 | 977.00 |
| **PBM Device** | 800.00 | - | 800.00 |
| **TOTAL** | | **21,164.56** | **13,637.46** |

**Table 2. Comparison of daily means over the entire follow-up period of progression in itens of the BATES-JENSEN scale.**

| BATES-JENSEN Scale Variable | Group | Mean* | Standard Deviation | Lower | Upper |
|---|---|---|---|---|---|
| **Total Score** | Control (sham) | 33,893 | 1,278 | 31,315 | 36,471 |
| | PBM | 34,564 | 1,640 | 31,255 | 37,872 |
| **Size** | Control (sham) | 2,637 | 0,142 | 2,351 | 2,923 |
| | PBM | 3,255 | 0,182 | 2,888 | 3,622 |
| **Type of Necrotic Tissue** | Control (sham) | 3,339 | 0,232 | 2,871 | 3,808 |
| | PBM | 2,564 | 0,298 | 1,963 | 3,165 |
| **Amount of Necrotic Tissue** | Control (sham) | 2,375 | 0,155 | 2,063 | 2,687 |
| | PBM | 1,819 | 0,198 | 1,419 | 2,219 |
| **Skin Color Around the Wound** | Control (sham) | 2,518 | 0,180 | 2,156 | 2,880 |
| | PBM | 1,559 | 0,230 | 1,094 | 2,024 |
| **Daily Mean Change in Scale Score** | Control (sham) | 1,200 | 1,300 | 2,156 | 2,880 |
| | PBM | 2,400 | 2,000 | 0,16 | 2,460 |

* Clinical outcomes of the study: Jana Neto FC, Martimbianco ALC, Mesquita-Ferrari RA, Bussadori SK, Alves GP, Almeida PVD, et al. Effects of multiwavelength photobiomodulation for the treatment of traumatic soft tissue injuries associated with bone fractures: A double-blind, randomized controlled clinical trial. J Biophotonics. 2023 May 1;16 (19).

The average duration of follow-up for the participants was 13.1 days (±11.5) in the PBM group and 23.1 days (±21.3) in the Control group. Nevertheless, this difference was not statistically significant between the two groups (p = 0.76), as already reported [19].

The total cost of treatment for the Control group was R$21,164.56, and a difference of R$7,527.10 more was observed when compared to the treatment of the PBM group. For the first analysis, the mean daily score of the BATES-JENSEN scale was considered the outcome measure. When calculating the cost-effectiveness ratio based on the daily average in the scale scores, it was observed that the cost-effectiveness ratio of the PBM group was lower; that is, R$424.71 were spent in the PBM group for each point reduced on average per day in the BATES-JENSEN scale. The cost-effectiveness ratio compared to the two groups was R$3,500.98 (negative). In addition to the difference observed in the clinical trial outcome, when considering this crude analysis, the PBM group was more cost-effective (spent less with the lower daily average on the scale). Table 4 represents the model adopted for the calculation of cost-effectiveness. The proposed intervention did not present incremental cost since the difference in the costs to reduce measures between the groups was smaller for the PBM group.

In a second economic analysis, the outcome was "wound resolution time" measured in days (Table 3). The mean time to resolution of the PBM group was 13.1 days, while for the Control group, it was 23.1 days, which confers an incremental effectiveness of 10 days for the PBM group. Regarding the cost-effectiveness ratio for this outcome, it was observed that the PBM

**Table 3. Cost-effectiveness analysis.**

| Analysis Type | Cost Measurement | Outcome Bates-Jensen | Ratio | Outcome Time to wound resolution | Ratio |
|---|---|---|---|---|---|
| Alternative | Cost | Effect | Cost/effect | Effect | Cost/effect |
| PBM Group | R$13,637.46 | 32.11 daily average on the BATES-JENSEN scale | R$424.71/per reduced point on the BATES-JENSEN scale | 13.1 days | Incremental cost = R$ -7527.1 |
| Control Group (CG) | R$ 21,164.56 | 34.26 daily averages on the BATES-JENSEN scale | R$617.76/per reduced point on the BATES-JENSEN scale | 23.1 days | Incremental effectiveness = -10 days R$ -7527.10 / -10 days = R$752.71 (negative) |

**Table 4. Cost-effectiveness matrix–average daily outcome BATES-Jensen scale.**

| Cost-effectiveness | Cost | Same cost | Higher cost |
|---|---|---|---|
| Lower effectiveness | A (performs ICER)] | B | C (Mastered) |
| Same effectiveness | D (Dominant) | E (arbitrary) | F |
| Higher effectiveness | G (PBM Group) | H | I (performs ICER) |

group was more cost-effective when compared to the total cost of treatment of the two groups and the time of wound resolution.

The ICER for this outcome was R$752.71 negative every ten days of wound resolution. It is assumed that if the entire sample had been treated with PBM, it would be possible to save R$7,527.10 and reduce the resolution time by ten days compared to the Control group (Table 3).

When analyzing the two outcomes, it was observed that there was a statistical difference in the evaluation of daily averages in the BATES-Jensen scale, and there was no statistical difference in the time of treatment. However, there was a difference in the costs of each group about the outcomes. Given this situation, the dominance of the treatments was analyzed through the cost-effectiveness matrix. The evaluation of the groups in the cost-effectiveness matrix compares costs and outcomes and the dominance of one over the other. When the procedure has lower cost and higher effectiveness than the other, it is in the dominant quadrant for it (G), that is, its use is the most indicated. The same indication happens when the procedure has the same effectiveness with a lower cost, or it has the same cost with higher effectiveness (D or E). However, when the two treatments have the same effectiveness and cost, the decision is arbitrary, and when a procedure has lower effectiveness and higher cost or vice versa (A or I), it is necessary to perform the ICER (Incremental Effectiveness Cost Ratio). In this case, the person who decides to choose the treatment needs to assess whether the additional value of the therapeutic alternative compensates for the clinical gain caused by the treatment. In the matrix, cells B, C, and F do not represent a new cost-effective alternative, while cells D, G, and H represent a new alternative to be considered. In this economic analysis, in the average daily score on the BJ scale, the PBM treatment was dominant because it presented greater effectiveness and lower cost. In contrast, resolution time, neither of the two treatments was dominant about the other, and the PBM, because it presented the same effectiveness and lower cost in relation to control group, is located in cell D, that is, an alternative to be considered for insertion in health services. Given this situation, the manager must clinically evaluate the clinical relevance and its costs for decision-making (Tables 4 and 5) [22].

## Discussion

The costs of fractures and orthopaedic surgeries are usually divided into direct and indirect costs, that is, those associated with treatment and those related to costs due to loss of productivity [9, 23]. In this study, the direct costs of treatment were evaluated, and the indirect costs were considered the days away from work of the research participants.

**Table 5. Cost-effectiveness matrix–time to resolution outcome.**

| Cost-effectiveness | Much Lower Cost | Same cost | Higher cost |
|---|---|---|---|
| Lower effectiveness | A (performs ICER)] | B | C (Mastered) |
| Same effectiveness | D (PBM Group) | E (arbitrary) | F |
| Higher effectiveness | G (Dominant) | H | I (performs ICER) |

In a systematic review published in 2021 [2] on the economic burden of tibial fractures, the authors reported that in middle and low-income countries, the most considerable financial losses are not related to the direct treatment costs but to the indirect cost due to workday losses. This analysis provides a compilation of relevant clinical information and treatment-derived costs. When comparing the two groups, the costs of changing dressings, days without work, and sick pay represented a more significant burden to the system.

In this context, the need to implement protocols in health services that can reduce the length of stay and the possibility of returning to productivity more quickly is emphasized. This study did not evaluate the secondary indirect costs related to the reallocation of family resources, the need to sell assets, and post-surgical rehabilitation treatment. However, the clinical result leads us to highlight the need for economic evaluation trials of these data related to PBM.

The treatment of tibial fractures is reported in the literature as a challenge due to complications, delay in resolution, and impact on the affected patient's life [24]. A cost analysis compared the treatment of tibial fractures with an autologous bone graft of the iliac crest (ICBG) using bone morphogenetic protein-7 (BMP-7). The ICBG was considered the control group, and the proposal with BMP-7 decreased the hospitalization time by two days with a higher average cost of 6.78% [24]. It is not possible to directly compare this information with the present study; however, the reduction in length of stay was considered relevant and was lower than that observed with PBM.

In this study, PBM was compared with a placebo, that is, an LED device was inserted as an adjuvant in standard hospital procedures, and even though it was an addition, it represented a 35% saving in the cost of general treatment due to the reduction in length of hospital stay [13–16, 19, 25]. As this is a preliminary study and presents limitations, we cannot yet extrapolate this economic result to all services. However, these results indicate that it may be interesting to develop a more comprehensive and complete study to prove such possibilities.

PBMs have been studied and considered effective in controlling pain, edema, and tissue repair A clinical trial with 68 participants with tibial fracture compared the use of laser in the wavelength of 830 nanometers and 26J/cm2 with a placebo. It demonstrated that the PBM group represented a faster resolution of pain and ambulation cases than the control [25].

The results of the randomized clinical trial that evaluated the effects of PBM in treating traumatic soft tissue injuries associated with bone fractures reaffirm the clinical importance of PBM therapy, its efficacy, and safety, with a reduction in the time required for definitive surgery and hospitalization [19].Furthermore, our results demonstrate the savings from the reduction in hospitalization time, suggesting that PBM can be considered a promising treatment with economic benefits for traumatic soft tissue injuries associated with lower limb fractures.

This scientific data was the first to economically evaluate using PBM as an adjuvant treatment in repairing soft tissue lesions after tibial fracture associated with soft tissue lesions. However, despite all the methodological efforts for clinical and economic analysis, because it is an area still under development in scientific production, it is necessary to increase the number of clinical trials for data composition and more complete and comprehensive economic analysis.

In the simulation of cost-effectiveness analysis in this study, the Incremental Cost-Effectiveness Ratio (ICER) matrix could observe a positive scenario for PBM insertion. Regarding the daily average of the BATES-Jensen scale, the experimental group showed higher effectiveness and lower cost; it would be classified as dominant in this economic analysis model. However, it is not yet possible to directly compare the literature with the existing data.

One of the limitations of this preliminary analysis is that this study does not provide a cost-utility analysis, considering the quality of life-adjusted life years (QALYs), which is regarded as an economically important aspect [9, 26, 27]. The present study can only be used to compare the insertion of pre-surgical ST in the Brazilian scenario with the usual protocol in the health system.

We admit that other factors need to be considered for further cost-effectiveness studies. Economic burdens with amputation, loss of permanent productivity, and other comorbidities can be inserted in future trials.

## Conclusion

This study found, in a preliminary way, that pre-surgical PBM in cases of tibial fracture was more effective in the BATES-Jensen Scale and had the same resolution time as the placebo. PBM presented a lower total cost in both outcomes than the control group. It is concluded, therefore, that PBM can be a supportive therapy of clinical and economic interest in the Brazilian health system.

## Supporting information

**S1 Checklist. *Plos One* clinical studies checklist.**
(DOCX)

## Author Contributions

**Conceptualization:** Frederico Carlos Jana Neto, Lara Jansiski Motta, Kristianne Porta Santos Fernandes.

**Data curation:** Frederico Carlos Jana Neto, Ana Luiza Cabrera Martimbianco, Lara Jansiski Motta, Kristianne Porta Santos Fernandes.

**Formal analysis:** Frederico Carlos Jana Neto, Ana Luiza Cabrera Martimbianco, Lara Jansiski Motta.

**Funding acquisition:** Frederico Carlos Jana Neto, Raquel Agnelli Mesquita-Ferrari, Sandra Kalil Bussadori, Estela Capelas Barbosa, Kristianne Porta Santos Fernandes.

**Investigation:** Frederico Carlos Jana Neto, Ana Luiza Cabrera Martimbianco, Diogo Valvano de Medeiros, Fernanda Carolina Felix, Lara Jansiski Motta, Kristianne Porta Santos Fernandes.

**Methodology:** Frederico Carlos Jana Neto, Ana Luiza Cabrera Martimbianco, Diogo Valvano de Medeiros, Fernanda Carolina Felix, Lara Jansiski Motta, Kristianne Porta Santos Fernandes.

**Project administration:** Frederico Carlos Jana Neto, Kristianne Porta Santos Fernandes.

**Resources:** Frederico Carlos Jana Neto, Raquel Agnelli Mesquita-Ferrari, Sandra Kalil Bussadori, Cinthya Cosme Gutierrez Duran, Estela Capelas Barbosa, Kristianne Porta Santos Fernandes.

**Software:** Diogo Valvano de Medeiros, Fernanda Carolina Felix.

**Visualization:** Ana Luiza Cabrera Martimbianco, Cinthya Cosme Gutierrez Duran, Lara Jansiski Motta, Estela Capelas Barbosa, Kristianne Porta Santos Fernandes.

**Writing – original draft:** Frederico Carlos Jana Neto, Raquel Agnelli Mesquita-Ferrari, Sandra Kalil Bussadori, Lara Jansiski Motta, Kristianne Porta Santos Fernandes.

**Writing – review & editing:** Frederico Carlos Jana Neto, Ana Luiza Cabrera Martimbianco, Cinthya Cosme Gutierrez Duran, Lara Jansiski Motta, Estela Capelas Barbosa, Kristianne Porta Santos Fernandes.

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
