## [Decision Letter · Decision Letter 0]

15 Aug 2023

PONE-D-23-17845Cost analysis of photobiomodulation in tibia fracture in the Brazilian public health systemPLOS ONE

Dear Dr. Fernandes,

Thank you for submitting your manuscript to PLOS ONE. After careful consideration, we feel that it has merit but does not fully meet PLOS ONE’s publication criteria as it currently stands. Therefore, we invite you to submit a revised version of the manuscript that addresses the points raised during the review process.

We look forward to receiving your revised manuscript.

Kind regards,

Melissa Orlandin Premaor, M.D., Ph.D

Academic Editor

PLOS ONE

Reviewers' comments:

Reviewer's Responses to Questions

**Comments to the Author**

1. Is the manuscript technically sound, and do the data support the conclusions?

Reviewer #1: Partly

Reviewer #2: Yes

2. Has the statistical analysis been performed appropriately and rigorously? 

Reviewer #1: No

Reviewer #2: Yes

3. Have the authors made all data underlying the findings in their manuscript fully available?

Reviewer #1: No

Reviewer #2: Yes

4. Is the manuscript presented in an intelligible fashion and written in standard English?

Reviewer #1: Yes

Reviewer #2: Yes

5. Review Comments to the Author

Reviewer #1: This manuscript has a lot of potential to be a top-tier publication and the subject area is very relevant. Unfortunately, the work has been poorly conducted. The methods and the result section need to be re-arranged and done to meet the minimum health economic standards see CHEERS checklist.

Background/ Introduction

There is more emphasis on the UK and US rather than Brazil where the research was conducted. I would suggest taking a more global approach. There a quite a number of systematic reviews that are related to this that can be cited.

Include a paragraph on economic evaluation/analysis relevant to the subject matter in the Background section.

Methods

In health economic analysis the term “cost-effectiveness” applies to the “Incremental cost-effectiveness ratio (ICER)”, in this study “cost-effectiveness” is being misused to mean “cost per effect”.

What are the characteristics or demographics of the sample population (i.e. the 27 participants)? This should be the first table.

The study did not mention the intervention cost. The intervention cost is what differentiates the intervention and control group costs. (See for example Cost-effectiveness Analysis of the Dental RECUR Pragmatic Randomized Controlled Trial: Evaluating a Goal-oriented Talking Intervention to Prevent Reoccurrence of Dental Caries in Children, https://link.springer.com/article/10.1007/s40258-022-00720-5)

Results

Table 1 represents the cost, is there a reference to the costs or are the hypothetical?

Tables 2, 3, 4, and 5 can be combined together (see table 4 of example paper: Cost-effectiveness of Memory Assessment Services for the diagnosis and early support of patients with dementia in England, https://journals.sagepub.com/doi/full/10.1177/1355819617714816; and table 3 of example paper: Cost-Effectiveness of Dementia Care Mapping in Care-Home Settings: Evaluation of a Randomised Controlled Trial, https://link.springer.com/article/10.1007/s40258-019-00531-1).

The term Effectiveness in the third column of Table 2 should be labelled “Effect”

No need to include formulas in the table. The methods section describes and references all types of formulas and methods.

Tables 6 and 7 can be combined if at all it needs to be there.

Discussion

The discussion will improve after the methods and result section are updated.

Reviewer #2: Dear authors! This investigated topic is very important, however, I have some questions that were not clear for my understanding, for example, was the control group matched? I saw that there were 27 participants in all. Did the inclusion criteria include similar ages and systemic health conditions? It would be interesting if in the abstract, methodology, these data were clarified for the reader. Likewise, in the discussion of the study, these data were presented and discussed, as it is known that both bone repair and soft tissue wounds are impacted by systemic conditions.

Another aspect I would like to point out is that the BATES-JENSEN scale should be presented to the reader, bearing in mind that this scale was the main criterion for defining the effectiveness of photobiomodulation therapy. Although previous work has been published, I think it is important to describe the photobiomodulation protocol in detail.

Reading the work, the question of the cost of the professional and the assistant team came up. Was it not possible to include this data? If it was not possible, wouldn't it be interesting to also discuss this aspect as a limitation of the study? I believe that only hospital costs were presented, but we know that there are often extra costs borne by patients. It would be interesting, therefore, to add this information.

6. PLOS authors have the option to publish the peer review history of their article (what does this mean?). If published, this will include your full peer review and any attached files.

Reviewer #1: No

Reviewer #2: No

---

## [Author Response · Author response to Decision Letter 0]

4 Sep 2023

Dear Editor

Melissa Orlandin Premaor

Initially, we thank you for taking the time to evaluate the article and for considering it for publication.

We have reviewed the points suggested by the reviewers.

The changes are in yellow in the text, and each point was answered for each reviewer.

Based on the evaluations received, we present below the changes made and their justifications:

Reviewer #1: This manuscript has a lot of potential to be a top-tier publication and the subject area is very relevant. Unfortunately, the work has been poorly conducted. The methods and the result section need to be re-arranged and done to meet the minimum health economic standards see CHEERS checklist.

Response: We appreciate your observation, and to improve the description in the methods and results sections, we followed the CHEERS checklist and adjusted the information in the article. Additionally, we have attached the checklist as supplementary material.

Background/ Introduction

There is more emphasis on the UK and US rather than Brazil where the research was conducted. I would suggest taking a more global approach. There a quite a few systematic reviews that are related to this that can be cited.

Include a paragraph on economic evaluation/analysis relevant to the subject matter in the Background section.

Response: We understand your perspective and agree. We have added more cost details and information from the systematic review by Schade A, Khatri C, Nwankwo H, Injury WC, 2021 undefined. The economic burden of open tibia fractures: A systematic review. Elsevier [Internet]. [cited 2022 Sep 1]; Available from: https://www.sciencedirect.com/science/article/pii/S002013832100125X.

Methods

In health economic analysis the term “cost-effectiveness” applies to the “Incremental cost-effectiveness ratio (ICER)”, in this study “cost-effectiveness” is being misused to mean “cost per effect”.

What are the characteristics or demographics of the sample population (i.e. the 27 participants)? This should be the first table.

The study did not mention the intervention cost. The intervention cost is what differentiates the intervention and control group costs. (See for example Cost-effectiveness Analysis of the Dental RECUR Pragmatic Randomized Controlled Trial: Evaluating a Goal-oriented Talking Intervention to Prevent Reoccurrence of Dental Caries in Children, https://link.springer.com/article/10.1007/s40258-022-00720-5)

Response: We appreciate the suggestion to adjust the terms. We have made the necessary changes, replacing the term "effectiveness" with "effect," as suggested. Regarding the cost of the intervention, as this is the first clinical trial conducted with photobiomodulation in this clinical condition, we do not have data on the cost of the procedure in the Brazilian healthcare system. Therefore, following the methodology of health economic analysis, we considered the cost of the equipment as the cost of the intervention since the only difference between the PBM and sham groups was the application of light. The equipment costs were added to the costs of the experimental group. We have made a modification in the article's wording to clarify this point. 

We have included the demographic information in the text and emphasized the reference to the clinical trial that presents the table of original clinical data (Jana Neto FC, Martimbianco ALC, Mesquita-Ferrari RA, Bussadori SK, Alves GP, Almeida PVD, et al. Effects of multiwavelength photobiomodulation for the treatment of traumatic soft tissue injuries associated with bone fractures: A double-blind, randomized controlled clinical trial. J Biophotonics. 2023 May 1;16(5). https://onlinelibrary.wiley.com/doi/10.1002/jbio.202200299)

Results

Table 1 represents the cost, is there a reference to the costs or are the hypothetical?

Tables 2, 3, 4, and 5 can be combined together (see table 4 of example paper: Cost-effectiveness of Memory Assessment Services for the diagnosis and early support of patients with dementia in England, https://journals.sagepub.com/doi/full/10.1177/1355819617714816; and table 3 of example paper: Cost-Effectiveness of Dementia Care Mapping in Care-Home Settings: Evaluation of a Randomised Controlled Trial, https://link.springer.com/article/10.1007/s40258-019-00531-1).

The term Effectiveness in the third column of Table 2 should be labelled “Effect”

No need to include formulas in the table. The methods section describes and references all types of formulas and methods.

Tables 6 and 7 can be combined if at all it needs to be there.

 Response: We agree with the reviewer's observation and, as suggested, we have removed the formulas and have left only the explanation of the analysis in the methodology section. We have grouped the data in the tables to make it clearer for the reader.

Discussion

The discussion will improve after the methods and result section are updated.

Response: We sought to improve the discussion following the new corrections and inclusions.

Reviewer #2: Dear authors! This investigated topic is very important, however, I have some questions that were not clear for my understanding, for example, was the control group matched? I saw that there were 27 participants in all. Did the inclusion criteria include similar ages and systemic health conditions? It would be interesting if in the abstract, methodology, these data were clarified for the reader. Likewise, in the discussion of the study, these data were presented and discussed, as it is known that both bone repair and soft tissue wounds are impacted by systemic conditions.

Response: Thank you for the feedback. To clarify the doubts and improve the description in the text of the studied sample, we have added details about the sample and the studied groups. The 27 participants were randomly divided into 2 study groups. One group received photobiomodulation therapy with 13 participants, while the group referred to as "sham" received the same procedures as the experimental group, but with the photobiomodulation equipment turned off. We have included the inclusion and exclusion criteria in the methods description. As the demographic data of the sample has already been published along with the clinical data, we have included this information in the text. We hope to have addressed this deficiency in the methodological description.

Another aspect I would like to point out is that the BATES-JENSEN scale should be presented to the reader, bearing in mind that this scale was the main criterion for defining the effectiveness of photobiomodulation therapy. Although previous work has been published, I think it is important to describe the photobiomodulation protocol in detail.

Response: We agree that the BATES-JANSEN scale is one of the main points of our assessment. Considering the observation, we have highlighted the description of the scale and its scoring in the methodology section. In the results section, we have also provided sample results based on the BATES-JENSEN scale. Table 2 contains the data we deem essential for conducting the economic analysis.

Reading the work, the question of the cost of the professional and the assistant team came up. Was it not possible to include this data? If it was not possible, wouldn't it be interesting to also discuss this aspect as a limitation of the study? I believe that only hospital costs were presented, but we know that there are often extra costs borne by patients. It would be interesting, therefore, to add this information.

Response: We agree with the reviewer that the costs of the professional team and patient-related costs are of utmost importance in economic evaluations; however, this is an initial study. It is not yet a procedure included in the Brazilian healthcare system's list of procedures, and therefore, it was not possible to gather this data. But we believe that this preliminary economic evaluation provides us with a foundation for developing the methodology for the next clinical trial, which will include assistant team costs and indirect costs in this treatment context. We have included this information as a study limitation.

---

## [Decision Letter · Decision Letter 1]

4 Oct 2023

PONE-D-23-17845R1Cost analysis of photobiomodulation in tibia fracture in the Brazilian public health systemPLOS ONE

Dear Dr Kristianne Fernandes,

Thank you for submitting your manuscript to PLOS ONE. After careful consideration, we feel that it has merit but does not fully meet PLOS ONE’s publication criteria as it currently stands. Therefore, we invite you to submit a revised version of the manuscript that addresses the points raised during the review process.

Please submit your revised manuscript by 4th of November. If you will need more time than this to complete your revisions, please reply to this message or contact the journal office at plosone@plos.org. Please include the following items when submitting your revised manuscript:A rebuttal letter that responds to each point raised by the academic editor and reviewer(s). You should upload this letter as a separate file labeled 'Response to Reviewers'.A marked-up copy of your manuscript that highlights changes made to the original version. You should upload this as a separate file labeled 'Revised Manuscript with Track Changes'.An unmarked version of your revised paper without tracked changes. You should upload this as a separate file labeled 'Manuscript'.If applicable, we recommend that you deposit your laboratory protocols in protocols.io to enhance the reproducibility of your results. Protocols.io assigns your protocol its own identifier (DOI) so that it can be cited independently in the future. For instructions see: https://journals.plos.org/plosone/s/submission-guidelines#loc-laboratory-protocols. Additionally, PLOS ONE offers an option for publishing peer-reviewed Lab Protocol articles, which describe protocols hosted on protocols.io. Read more information on sharing protocols at https://plos.org/protocols?utm_medium=editorial-email&utm_source=authorletters&utm_campaign=protocols.

We look forward to receiving your revised manuscript.

Kind regards,

Melissa Orlandin Premaor, M.D., Ph.D

Academic Editor

PLOS ONE

Journal Requirements:

Reviewers' comments:

Reviewer's Responses to Questions

**Comments to the Author**

1. If the authors have adequately addressed your comments raised in a previous round of review and you feel that this manuscript is now acceptable for publication, you may indicate that here to bypass the “Comments to the Author” section, enter your conflict of interest statement in the “Confidential to Editor” section, and submit your "Accept" recommendation.

Reviewer #1: All comments have been addressed

Reviewer #2: All comments have been addressed

2. Is the manuscript technically sound, and do the data support the conclusions?

Reviewer #1: Yes

Reviewer #2: Yes

3. Has the statistical analysis been performed appropriately and rigorously? 

Reviewer #1: (No Response)

Reviewer #2: Yes

4. Have the authors made all data underlying the findings in their manuscript fully available?

Reviewer #1: Yes

Reviewer #2: Yes

5. Is the manuscript presented in an intelligible fashion and written in standard English?

Reviewer #1: Yes

Reviewer #2: Yes

6. Review Comments to the Author

Reviewer #1: **Table 3: The first column with the Ratio should differ from the second column with the Ratio. The second column ratio should be indicated as ICER (Incremental cost-effectiveness ratio)**.

**In the last row last column of Table 3, what does (negative) mean?? This is ambiguous and should be taken out.**

**In the text the word “negative” should be removed e.g “The cost-effectiveness ratio compared to the two groups was R$3,500.98 (negative)”. This sentence does not support or negate any intervention, does it mean the PMB group was better, or was the control better? All economic evaluations are comparisons of at least two interventions. This word negative was used in 6 (six) other places, can they be removed?**

Reviewer #2: Dear authors, I think that the work has now been improved with the information that was suggested. As you highlighted that it is a preliminary study, my suggestion is that you continue to investigate the cost-benefit and effectiveness of this treatment modality and that it can actually be incorporated into the public health system of different countries. Mainly from countries that require scientific evidence for the incorporation of therapeutic protocols and/or new healthcare technologies.

7. PLOS authors have the option to publish the peer review history of their article (what does this mean?). If published, this will include your full peer review and any attached files.

Reviewer #1: No

Reviewer #2: No

---

## [Author Response · Author response to Decision Letter 1]

26 Oct 2023

Dear Reviewers,

First and foremost, we would like to express our sincere gratitude for your invaluable time. Please find below the responses to your queries and suggestions:

Reviewer #1: Table 3: The first column with the Ratio should differ from the second column with the Ratio. The second column ratio should be indicated as ICER (Incremental cost-effectiveness ratio).

In the last row last column of Table 3, what does (negative) mean?? This is ambiguous and should be taken out.

In the text the word “negative” should be removed e.g “The cost-effectiveness ratio compared to the two groups was R$3,500.98 (negative)”. This sentence does not support or negate any intervention, does it mean the PMB group was better, or was the control better? All economic evaluations are comparisons of at least two interventions. This word negative was used in 6 (six) other places, can they be removed?

Response to Reviewer: We appreciate the suggestion for the correction in Table 3 regarding the columns of ratio and ICER. We have made the correction to the second column of ratio as suggested.

We agree that the word "negative" may introduce difficulties in comprehension of the analysis. We have removed it from Table 3 and all other instances where it appeared. 

By removing the word "negative," we reformulated the text to explain that the term "negative" signifies a savings of R$3,500.98 when employing the PBM technique. Thus, in our analysis, PBM outperformed the control group.

We've marked them in yellow for your convenience. Additionally, you can also find the changes outlined below:

The cost-effectiveness index between the two groups was R$3,500.98, indicating a savings of R$3,500.98 with the PBM technique. In addition to the observed difference in the clinical trial results, considering this simplified analysis, the PBM group demonstrated greater cost-effectiveness, spending less with a lower daily average on the scale. Table 4 illustrates the model used for the calculation of cost-effectiveness. The proposed intervention did not incur additional costs, as the disparity in costs for reducing measurements between the groups was smaller for the PBM group.

The ICER for this outcome represented a savings of R$752.71 for every ten days per resolved wound. It is assumed that if the entire sample had been treated with PBM, it would be possible to save R$7,527.10 and reduce the resolution time by ten days compared to the Control group (Table 3).

Reviewer #2: Dear authors, I think that the work has now been improved with the information that was suggested. As you highlighted that it is a preliminary study, my suggestion is that you continue to investigate the cost-benefit and effectiveness of this treatment modality and that it can actually be incorporated into the public health system of different countries. Mainly from countries that require scientific evidence for the incorporation of therapeutic protocols and/or new healthcare technologies.

Response to Reviewer: We sincerely appreciate your valuable suggestion. Indeed, it is highly pertinent. We will certainly continue our investigations into the cost-benefit and effectiveness of this treatment modality. We are committed to assessing its feasibility for integration into the public health system. Thank you for your insightful guidance.

We appreciate your valuable time to evaluate our work and your valuable suggestions. 

We hope the article aligns with your expectations with the changes made.

Thank you once again for your consideration.

Sincerely,

Authors

---

## [Editor Report · Decision Letter 2]

30 Oct 2023

Cost analysis of photobiomodulation in tibia fracture in the Brazilian public health system

PONE-D-23-17845R2

Dear Dr. Barbosa,

We’re pleased to inform you that your manuscript has been judged scientifically suitable for publication and will be formally accepted for publication once it meets all outstanding technical requirements.

Kind regards,

Michael R Hamblin

Academic Editor

PLOS ONE
---

## [Editor Report · Acceptance letter]

30 Nov 2023

PONE-D-23-17845R2 

Cost analysis of photobiomodulation in tibia fracture in the Brazilian public health system 

Dear Dr. Barbosa:

I'm pleased to inform you that your manuscript has been deemed suitable for publication in PLOS ONE. Congratulations! Your manuscript is now with our production department. 

Kind regards, 

on behalf of

Dr. Michael R Hamblin 

Academic Editor

PLOS ONE